# Elovanoids Counteract Inflammatory Signaling, Autophagy, Endoplasmic Reticulum Stress, and Senescence Gene Programming in Human Nasal Epithelial Cells Exposed to Allergens

**DOI:** 10.3390/pharmaceutics14010113

**Published:** 2022-01-04

**Authors:** Alfredo Resano, Surjyadipta Bhattacharjee, Miguel Barajas, Khanh V. Do, Roberto Aguado-Jiménez, David Rodríguez, Ricardo Palacios, Nicolás G. Bazán

**Affiliations:** 1Department of Health Science, Public University of Navarra, 31006 Pamplona, Spain; roberto.aguado@unavarra.es; 2Neuroscience Center of Excellence, Louisiana State University Health New Orleans (LSUHSC), New Orleans, LA 70112, USA; bhattacharjee@lsuhsc.edu (S.B.); kdo@lsuhsc.edu (K.V.D.); 3Diater Laboratorio, 28919 Madrid, Spain; d.rodriguez@diater.com

**Keywords:** elovanoids, allergy, house dust mite, inflammation, anti-inflammatory, lipid mediators

## Abstract

To contribute to further understanding the cellular and molecular complexities of inflammatory-immune responses in allergic disorders, we have tested the pro-homeostatic elovanoids (ELV) in human nasal epithelial cells (HNEpC) in culture challenged by several allergens. ELV are novel bioactive lipid mediators synthesized from the omega-3 very-long-chain polyunsaturated fatty acids (VLC-PUFA,n-3). We ask if: (a) several critical signaling events that sustain the integrity of the human nasal epithelium and other organ barriers are perturbed by house dust mites (HDM) and other allergens, and (b) if ELV would participate in beneficially modulating these events. HDM is a prevalent indoor allergen that frequently causes allergic respiratory diseases, including allergic rhinitis and allergic asthma, in HDM-sensitized individuals. Our study used HNEpC as an in vitro model to study the effects of ELV in counteracting HDM sensitization resulting in inflammation, endoplasmic reticulum (ER) stress, autophagy, and senescence. HNEpC were challenged with the following allergy inducers: LPS, poly(I:C), or *Dermatophagoides farinae* plus *Dermatophagoides pteronyssinus* extract (HDM) (30 µg/mL), with either phosphate-buffered saline (PBS) (vehicle) or ELVN-34 (500 nM). Results show that ELVN-34 promotes cell viability and reduces cytotoxicity upon HDM sensitization of HNEpC. This lipid mediator remarkably reduces the abundance of pro-inflammatory cytokines and chemokines IL-1β, IL-8, VEGF, IL-6, CXCL1, CCL2, and cell adhesion molecule ICAM1 and restores the levels of the pleiotropic anti-inflammatory IL-10. ELVN-34 also lessens the expression of senescence gene programming as well as of gene transcription engaged in pro-inflammatory responses. Our data also uncovered that HDM triggered the expression of key genes that drive autophagy, unfolded protein response (UPR), and matrix metalloproteinases (MMP). ELVN-34 has been shown to counteract these effects effectively. Together, our data reveal a novel, pro-homeostatic, cell-protective lipid-signaling mechanism in HNEpC as potential therapeutic targets for allergies.

## 1. Introduction

The challenge to effectively manage allergic rhinitis, asthma, and other allergic conditions is due in part to our limited understanding of the mechanisms involved. Damage to cell barrier integrity is part of the pathogenesis. House dust mites (HDM) are the most perennial indoor allergen source and a driving factor of underlying allergic rhinitis, allergic asthma as a result of the activation of chronic inflammatory airways, airway obstruction, and airway hyperresponsiveness (AHR). Approximately 1–2% of the global population suffers from HDM allergy and sensitization, leading to chronic allergic diseases, including allergic rhinitis, asthma, and atopic dermatitis. Asthma is caused by exposure to aeroallergens such as dust mites, pet dander, pollen, or mold. Of these allergens, HDM are the most extensive indoor allergen, and they comprise *Dermatophagoides pteronyssinus* and *Dermatophagoides farinae* (in dry areas) and *Euroglyphus maynei* (in humid areas) [1,2,3,4,5,6,7,8,9,10].

HDM allergen triggers allergic inflammation through the activation of the innate immune Toll-like receptors (TLR), protease-activated receptors (PAR), and dendritic-cell-specific intercellular adhesion molecule-3-grabbing non-integrin (DC-SIGN), which are activated by HDM allergens, resulting in the upregulation of pro-inflammatory cytokines [6,7,11,12,13,14]. Exposure of airway epithelial cells to HDM activates NF-κB pathways within the airway epithelium and contributes to allergic inflammation, AHR, and fibrotic airway remodeling [15].

Elovanoids (ELV) are bioactive lipid mediators derived from very-long-chain polyunsaturated fatty acids (VLC-PUFA,n-3), which are the biosynthetic products of the newly discovered elongase ELOVL4 (Appendix A). These novel lipid mediators are able to downregulate senescence gene programming, senescence-associated secretory phenotype (SASP), and inflammaging (age-dependent, low-grade, chronic inflammation), which in turn promotes and sustains cell structural integrity, viability, and function [16,17,18,19,20].

Because LCPUFAs have been shown to play an important role in the control of mediators involved in inflammation, exerting a protective role of cellular integrity against inflammation [6,7,8,21,22], we proposed to determine whether ELV could be useful to protect human nasal epithelial cells (HNEpC) exposed to HDM-induced deleterious actions. Moreover, it is well established that several bioactive metabolites synthesized from the omega-3 VLC-PUFA,n-3 counter regulate airway inflammation in asthma [23,24].

Some serum markers of inflammation (e.g., IL-6) tend to be slightly higher in older adults than the usual baseline in younger adults [25]. It has been proposed that this causes damage to normal tissues and contributes to degenerative diseases of aging or inflammaging phenomenon [26,27,28]. The inflammaging is partly attributable to dysregulated immune systems leading to the development of the SASP program that consists of the release of multiple cytokines, chemokines, and mitokines by the accumulation of senescent non-immune tissue cells with a pro-inflammatory effect [29]. This dysregulation has been associated with environmental allergens or pollutants not only in the SASP program but also in the processes of autophagy and mitophagy [30].

Therefore, to contribute to further understanding of the cellular and molecular complexities of inflammatory-immune responses in allergic disorders, we have tested different mechanisms by which aeroallergens trigger pathology with the idea in mind to define targets and mediators that could counteract allergic responses and demonstrate the role that ELV play in the expression of several key genes in HDM-induced HNEpC. The selected genes are related to cellular processes, such as autophagy [31,32,33,34], the unfolded protein response (UPR) [35], matrix metalloproteinases (MMP) [36], as well as markers of senescence and inflammation. The results obtained from this study will make it possible to determine a new pro-homeostatic and cell-protective lipid signaling mechanism in nasal epithelial cells with the aim of selecting new potential therapeutic targets for the treatment of allergies.

## 2. Materials and Methods

### 2.1. Experimental Design

The experimental design (Figure 1) with primary HNEpC allowed us to assess the pro-homeostatic protective signaling of ELVN-34 in relation to cytotoxicity, cell viability, cytokine production, and expression of genes involved in inflammatory and senescence processes. To test the efficacy of ELV on cell survival, HNEpC were challenged with different stressors/allergy inducers—lipopolysaccharide (LPS), poly(I:C), or house dust mite extracts (HDM) (30 μg/mL) and, 30 min later, the cells were treated with ELVN-34 (500 nM) and incubated for 24 h. LPS is the principal component of Gram-negative bacteria that activates the innate immune system through its recognition by Toll-like receptor 4 (TLR4). This leads to a signaling cascade that ultimately results in the activation of NF-κB and the production of pro-inflammatory cytokines [37,38,39]. Polyinosinic-polycytidylic acid (poly(I:C) or poly(rI):poly(rC)) is a synthetic analog of double-stranded viral RNA (dsRNA), a molecular pattern associated with a viral infection, such as loss of epithelial integrity, increased production of mucus, and inflammatory cytokines [40]. Poly(I:C), a TLR3 agonist, activates the antiviral pattern recognition receptors TLR3, RIG-I/MDA5, and PKR [41,42], thereby inducing signaling via multiple inflammatory pathways, including NF-κB and IRF [43,44]. High molecular weight poly(I:C) comprises long strands of inosine poly(I) homopolymer annealed to strands of cytidine poly(C) homopolymer. The average size of Poly(I:C) HMW is from 1.5 kb to 8 kb.

Cells were analyzed for cytotoxicity and cell viability, and cell culture supernatants were also collected at 24 h to assay the expression of pro-inflammatory and anti-inflammatory cytokines, chemokines, and cell adhesion molecule (ICAM1) by quantitative ELISAs. HNEpC were incubated with allergy inducers (30 μg/mL), and 30 min later, ELVN-34 (500 nM) were added and incubated for 24 h (Figure 1).

### 2.2. Cell Line

Cryopreserved HNEpC were purchased from PromoCell GmbH, Heidelberg, Germany (Catalog# C-1260, Lot# 436Z028). These cells are primary nasal epithelial cells obtained from the nasal mucosa of a 50-year-old Caucasian male. HNEpC were grown to 80% confluence in PromoCell’s Airway Epithelial Cell Growth Medium (Catalog# C-21060), which was supplemented with Airway Epithelial Cell Growth Medium SupplementPack (Catalog# C-39160) and Penicillin/Streptomycin.

### 2.3. Inducers

HNEpC were challenged using several stressors: House dust mite (HDM) extract from *Dermatophagoides pteronyssinus* (D.P.) (Catalog# 3033) and from *Dermatophagoides farinae* (D.F.) (Catalog# 3040)—lyophilized extract was obtained from Chondrex, Inc. (Woodinville, WA, USA) and used at 30 μg/mL to challenge HNEpC. To investigate whether there were any synergic effects of D.P. and D.F., a mixture of both D.P. and D.F. was used at 15 μg/mL + 15 μg/mL to challenge HNEpC. Lipopolysaccharide (LPS) from *Escherichia coli* serotype 0111:B4 was used (Sigma-Aldrich Catalog# L4391). LPS we used for our experiments is a preparation of smooth (S)-form LPS purified from the Gram-negative *E. coli* 0111:B4 that was used at 30 μg/mL to challenge HNEpC. Poly(I:C) (Catalog# P1530) were obtained from Sigma-Aldrich and used at 30 μg/mL to challenge HNEpC.

### 2.4. Cell Survival Assay

Cell viability was analyzed using PrestoBlue HS (ThermoFisher Scientific, Waltham, MA, USA) reagent (Catalog# C50201, Invitrogen). The PrestoBlue HS Cell Viability Reagent is a complete nontoxic add-and-read reagent that does not require cell lysis. The highly purified resazurin used for PrestoBlue HS results in a reagent with >50% decrease in background fluorescence and >100% increase in the signal-to-background ratio.

On entering live cells, the cellular reducing environment reduces resazurin to resorufin, a red and highly fluorescent compound. Viable cells continuously convert resazurin to resorufin, increasing the overall fluorescence and color of the media surrounding the cells. Moreover, the conversion of resazurin to resorufin results in a pronounced color change. Therefore, cell viability can be detected using absorbance-based plate readers. Fluorescence is read using a fluorescence excitation wavelength of 560 nm (excitation range is 540–570 nm) and an emission of 590 nm (emission range is 580–610 nm).

### 2.5. Cytotoxicity Analysis

Cytotoxicity was measured using CyQuant LDH assay kit (Catalog# C20301, ThermoFisher Scientific), which evaluates damage to the plasma membrane that releases LDH into the surrounding cell culture media. The extracellular LDH in the media can be quantified by a coupled enzymatic reaction in which LDH catalyzes the conversion of lactate to pyruvate via NAD^+^ reduction to NADH. Oxidation of NADH by diaphorase leads to the reduction of a tetrazolium salt (INT) to a red formazan product that can be measured spectrophotometrically at 490 nm. The level of formazan formation is directly proportional to the amount of LDH released into the medium, which is indicative of cytotoxicity.

### 2.6. Pro-Inflammatory/Anti-Inflammatory Cytokine Profile and Cell Adhesion Analysis

In addition, the expression of the following cytokines was analyzed in the supernatant of HNEpC exposed to different allergic inductors: IL-6 (Catalog# 6802, Chondrex) range of detection: (9–600 pg/mL), IL-1β (Catalog# 6805, Chondrex) range of detection: (4–250 pg/mL) IL-8/CXCL8 (Quantikine ELISA Kit, Catalog# D8000C, R&D Systems) range of detection: (31–2000 pg/mL), vascular endothelial growth factor (VEGF) (Catalog# 6810, Chondrex) range of detection: (31–2000 pg/mL), ICAM1(CD54) (ELISA Kit, Catalog# ab100640, Abcam) range of detection: (16–1000 pg/mL), CXCL1/KC/GRO (Catalog# 6825, Chondrex) range of detection: CCL2/MCP-1 (Catalog# 6821, Chondrex) range of detection: (16–1000 pg/mL), and IL-10 (Catalog# 6806, Chondrex) range of detection: (8–500 pg/mL). The standard curves for each of the measured markers are shown in Appendix A.

### 2.7. Cellular Senescence Assay (β-Galactosidase Expression)

Cellular senescence was measured in HNEpC by analyzing the expression of beta-galactosidase using a beta-gal staining kit (Spider β-gal Kit, Catalog#SG04, Dojindo). SPiDER-β-Gal allows the detection of senescence-associated (SA)-β-gal with high sensitivity, being a reagent that possesses high cell permeability and high retentivity inside cells. SA-β-gal can be detected not only in living cells but also in fixed cells using a reagent (Bafilomycin A1) to inhibit endogenous β-gal activity.

### 2.8. Gene Expression

Total RNA was isolated using RNeasy Plus Mini Kit (Qiagen, Hilden, Germany). Quantitative PCR was performed in a CFX-384 Real-Time PCR system (Bio-Rad, Hercules, CA, USA). The expression of target genes was normalized to the geometric mean of housekeeping genes (ACTB and GAPDH), and relative expression was calculated by the comparative threshold cycle method (ΔΔCT). Appendix A shows a list of the primers used in the study.

### 2.9. Statistical Analysis

All the values in the figures and text are presented as mean ± SEM of n observations. Statistical analyses were performed using GraphPad Prism 9.2.0. (GraphPad, La Jolla, CA, USA). A one-way ANOVA was performed when comparing multiple groups with the appropriate Holm–Sidak’s post hoc test. When data were found not to be normally distributed, the Mann–Whitney test was used when comparing two groups and a Kruskal–Wallis test when comparing multiple groups. All treatments were compared to relevant vehicle control groups, and differences were deemed significant when *p* < 0.05.

## 3. Results

### 3.1. ELV Elicit Potent Cytoprotection against LPS, Poly(I:C), or HDM

LPS and poly(I:C) induced a decrease in cell viability and an increase of LDH levels (Figure 2, red bars), indicating that HNEpC were subjected to cellular stress that compromises their viability. Our results indicate that ELVN-34 successfully prevented cell death, restoring the cell viability (Presto Blue HS assay) (Figure 2, left panel) and LDH levels (cytotoxicity assay, CyQuant LDH) (Figure 2, right panel) in all the conditions tested. The addition of ELV increases cell viability and fosters protection to the HNEpC.

*D. farinae*, *D. pteronyssinus*, and HDM induced a decrease in cell viability (Figure 2, red bars at left panel) and increased LDH levels (Figure 2, red bars at right panel), indicating that HNEpC were subjected to cellular stress that compromises their viability. Cell viability assay using Presto Blue HS reagent also shows more resazurin to resorufin production in untreated cells or cells challenged with the inducers (Figure 2, left panel) and treated with ELVN-34.

Cytotoxicity assay (LDH) shows that the inducers—LPS, polyI:C, *D. farinae*, *D. pteronyssinus*, and HDM—increase the formation of red formazan, indicating cytotoxicity, which is reduced by ELVN-34 (Figure 2, right panel).

Our results indicate that ELVN-34 successfully prevented cell death, restoring the cell viability (Presto Blue HS assay) (Figure 2, left panel) and LDH levels (cytotoxicity assay, CyQuant LDH) (Figure 2, right panel) after the challenge with different HDM extracts. In summary, the addition of ELV increases cell viability as well as HNEpC protection.

### 3.2. ELV Downregulate the Expression of Pro-Inflammatory Cytokines and Chemokines Induced by LPS, Poly(I:C), or HDM in HNEpC

HNEpC were exposed 30 min to LPS, or poly(I:C) or HDM (30 μg/mL), and their supernatants were collected in order to determine the expression of pro-inflammatory cytokines and chemokines IL-6, IL-1β, IL-8/CXCL8, CCL2/MCP-1, CXCL1/KC/GRO, VEGF, and cell adhesion molecule ICAM1(CD54). Our results demonstrate that the challenge of HNEpC with LPS or poly(I:C) induces the expression of pro-inflammatory cytokines and chemokines IL-6, IL-1β, IL-8/CXCL8, CCL2/MCP-1, CXCL1/KC/GRO, VEGF, and ICAM1(CD54) (Figure 3, red bars). The treatment of HNEpC with ELVN-34 (500 nM) reduces the expression of pro-inflammatory cytokines and chemokines (Figure 3). However, the allergy inducers—LPS and poly(I:C)—downregulate the expression of anti-inflammatory IL-10 in HNEpC that is restored to normal levels after treatment with ELVN-34 (Figure 3, panel IL-10). These results show that treatment with ELV protects HNEpC from the expression of pro-inflammatory cytokines under the stressors LPS and poly(I:C), favoring the expression of anti-inflammatory cytokines such as IL-10.

Similar results were observed when HNEpC were challenged with *D. farinae*, *D. pteronyssinus*, and (HDM) (30 μg/mL) (Figure 4, red bars). The treatment of HNEpC with ELVN-34 (500 nM) reduces the expression of pro-inflammatory cytokines and chemokines induced by HDM stressors (Figure 4). However, for ICAM1 (CD54) upregulation in the presence of HDM, though there was downregulation upon ELVN-34 treatment, yet it did not reach statistical significance. Again, HDM showed significant downregulation of the IL-10 expression in HNEpC that is restored to normal levels after treatment with ELVN-34 (Figure 4, panel IL-10).

These results indicate that treatment with ELV protects HNEpC from the expression of pro-inflammatory cytokines in favor of anti-inflammatory cytokines, suggesting a possible beneficial therapeutic effect for the treatment of allergies.

### 3.3. ELV Protect HNEpC from Senescence

Finally, our study evaluated the SASP response in HNEpC, finding that treatment with ELV protected these cells from this phenomenon. HDM (30 μg/mL) induced an increased number of beta-galactosidase-positive cells (Figure 5, red bar), indicating that cellular senescence has been activated as a consequence of the stressor used. ELVN-34 (500 nM) also elicits robust protection of HNEpC exposed to HDM extract (Figure 5, green bar), suggesting that this novel compound could protect HNEpC from premature senescence induced by the exposure to allergens.

### 3.4. ELV Regulate the Expression of Autophagy, UPR, MMP, Senescence, and Inflammation Markers in HNEpC

To search for the mechanism involved in ELV protection against HDM-mediated damage, UPR, MMP, senescence, and inflammation markers were analyzed by quantitative PCR on HNEpC challenged with *D. farinae*, *D. pteronyssinus*, and HDM (30 μg/mL). We selected genes involved in autophagy (ATG3, ATG5, ATG7, BECLIN-1, and P62) and UPR pathways (ATF6, CHOP, and IRE1) involved in endoplasmic reticulum (ER) stress, and MMP genes (MMP8, MMP2, MMP9, MMP3, MMP12, TIMP1, and TIMP2) as well as the expression of senescence (P21, P16, P27, and P53) and inflammation genes (IL-1α, IL-6, and IL-1β) (Figure 6). All the markers analyzed showed high expression in HNEpC after HDM sensitization, with the exception of p53 (Figure 6, red bars). The treatment of HNEpC with ELVN-34 (500 nM) significantly reduces the expression of all genes mentioned above after challenge with HDM (Figure 6, blue bars). Moreover, when HNEpC were treated with ELVN-34 only (without HDM stimulus), there were no significant changes in expression of these genes (Figure 6, green bars) in comparison to untreated cells (baseline levels) (Figure 6, black bars).

These results clearly indicate that ELV have a protective effect on survival, integrity, senescence, and inflammation, as well as the autophagy process at the cellular level.

## 4. Discussion

Cellular and molecular events that trigger and sustain inflammatory-immune responses and trigger and sustain allergic disorders are recapitulated in primary HNEpC exposed to inflammatory inducers, LPS and poly(I:C), or HDM. Both increased cytotoxicity (damage to the plasma cell membrane) and cell viability (metabolic activity of cells) do occur upon exposure to these known inflammatory inducers of innate immunity, LPS [38] and poly(I:C) [45,46], as well as against HDM *Dermatophagoides pteronyssinus* and *Dermatophagoides farinae*.

The high sensitization capacity of HDM induces proliferation of Th0 to Th2 lymphocytes, limiting the differentiation to Th1. It has also been attributed to the ability of group 1 allergens (Der p1), for example, to cleave IL-2R [47], activate dendritic cells for Th2 polarization, suppress IL-12 by CD40 cleavage, reduce INF-γ release, and increase IL-4 production [48], thus fostering an allergic microenvironment by acting on adaptive immunity. However, there is growing evidence that these group 1 allergens are mainly responsible for activating the innate immune response, cleaving the binding proteins of epithelial cells, activating protease-sensitive receptors (PAR) and TLR4, and activating NOD-like receptors. HDM *Dermatophagoides pteronyssinus* activated eosinophils and bronchial epithelium cells, expressed increased release of inflammatory cytokines (NF-κB, AP-1, MAPK, IL-1, IL-6, and TNF-α), and enhanced expression of the adhesion molecule ICAM-1 [49]. In our cellular model of HNEpC, we have confirmed the increased abundance of pro-inflammatory cytokines of epithelial origin and alarmins, such as IL-6, IL-8, and IL-1α and IL-1β, as well as the pro-inflammatory chemokine CCL2 associated with Th2 response and monocyte chemotactic, and the pro-inflammatory chemokine CXCL1, a potent chemotactic for neutrophils.

Bronchial epithelial cells act as immunoregulators in allergic respiratory diseases through the expression of adhesion molecules on the cell surface, such as ICAM1 [50,51]. We have also confirmed the HDM-induced expression of vascular endothelial growth factor (VEGF) and intercellular adhesion molecule of endothelial and immune cells (ICAM1). Although the increase in angiogenesis is a feature of the remodeling of the asthmatic airways as a consequence of inflammation, its activation takes place from the beginning of exposure to the allergen, associated with an expression of ICAM-1 molecules that will favor the adhesion of inflammatory effector cells.

IL-10 is expressed in the nasal mucosa of allergic patients both in the epithelium and in the blood vessel endothelial cells, playing a role in the pathogenesis of allergic rhinitis as an immunosuppressive cytosine and revealed by Muller et al. [52] after nasal challenges with allergens and subsequent biopsies. Similarly, in our experimental model, we have observed a marked decrease in the production of IL-10 (an anti-inflammatory cytokine) after induction with LPS, poly(I:C), and HDM.

On the other hand, autophagy plays an essential role in the lung inflammatory response as well as in the pathogenesis of asthma [53]. High levels of autophagy have been seen in sputum granulocytes, peripheral blood cells, and eosinophils from patients with asthma, although little is known about their role in allergic rhinitis. Moreover, genetic mutations in autophagy genes have been associated with asthma; for example, polymorphisms in ATG5 correlate with airway remodeling and loss of lung function in asthma [33]. Environmental pollutants and allergens can induce autophagy, and autophagy has been suggested to be essential for inhibiting spontaneous lung inflammation and secretion of airway mucus by goblet cells. Mice with autophagy deficiency (Atg5^−/−^ and Atg7^−/−^) develop spontaneous sterile lung inflammation [31]. Cell culture experimental model systems suggest that, in the lung epithelium, activation of autophagy may be deleterious, inducing disease progression and promoting the Th2 epithelial response in asthma. Our data also show that HDM triggered the expression of several key genes in the nasal epithelium, including autophagy, unfolded protein response, matrix metalloproteinases, senescence, and inflammation [30]. HDM activates the autophagy genes: ATG3, ATG5, ATG7, BECLIN-1, and P62. These data support other previous studies in other respiratory diseases, as well as in chronic obstructive pulmonary disease (COPD) [34,53].

ER stress and the UPR pathway may play a fundamental role in the regulation of epithelium-dendritic cell interaction, which is crucial for the initiation and amplification of the allergic response [54,55]. The accumulation of misfolded proteins in the ER can be signaled through IRE1 (transmembrane receptor which includes the enzyme 1 that requires inositol), ATF6 (activation of transcription factor 6), and double-stranded RNA-dependent protein kinase PKR—ER kinase (PERK). If the first two signaling pathways fail to restore ER stress initiated by the GRP78 chaperone, C/EBP homologous protein (CHOP) will mediate apoptosis using PERK signaling. In our cell model, treatment with HDM also increases the expression of UPR, including the ATF6, CHOP, and IRE1 genes, which are critical in allergic reactions and in asthmatic patients [35,55].

Cellular senescence is characterized by irreversible cell cycle arrest, leading to an increase in pro-inflammatory mediators and less capacity for tissue regeneration or repair, greater accumulation of senescent cells [56] being an important risk factor in the lung for the development of asthma [57]. Among the studies that relate cell senescence and asthma, it is worth mentioning the research performed by Wu et al. [58], who observed cellular senescence induced by Thymic Stromal Lymphopoietin (TSLP) with elevated levels of P21 and P16 in human epithelial cells, being essential for the remodeling of the airways in vitro; as well as the study with a plasminogen activator inhibitor (PAI-1), a mediator of cell senescence and fibrosis, which could activate p53 and mediate the senescence of type II alveolar cells (ATII) induced by bleomycin and doxorubicin [59]. Therefore, senescence is triggered by factors such as aging, DNA damage, oxidative stress, and mitochondrial dysfunction [30]. In order to study the activation of the cell senescence pathways, we can use different markers: the p53-ARF pathway (crucial mediator that plays a role in the cellular responses to DNA damage) and the p16-pRb pathway (induced by a variety of stressful stimuli). In our results, the inflammatory genes IL-1α, IL-1β, and IL-6 are also activated, and finally, HDM also triggered senescence genes, including Cdkn1a (p21), Cdkn2a (p16), and Cdkn1b (p27), but not p53.

In an experimental model of retinal cells, ELV display protection against neurodegenerative diseases resembling age-related macular degeneration and Alzheimer’s disease, promoting repair, remodeling, and regeneration [18]. ELVN-34 promoted cell viability and decreasing cytotoxicity after HNEpC sensitization to HDM. Furthermore, our results indicate that the levels of IL-6, IL-1β, IL-8, VEGF, ICAM1, CXCL1, and CCL2, which had increased with HDM sensitization, were significantly reduced with ELVN-34 treatment. Likewise, the level of IL-10 was restored by treatment with ELVN-34. Therefore, ELVN-34 manages to decrease the production of pro-inflammatory cytokines, increasing the production of anti-inflammatory cytokines and decreasing the expression of adhesion molecules and vascular endothelial growth factor after exposure to inflammatory inducers and the HDM allergen.

Liu et al. [32], reviewing the role of autophagy in allergic inflammation, observed that inhibition of autophagy in a murine OVA asthma model reduces airway response, eosinophilia, and inflammation. Interestingly, treatment with ELVN-34 decreases the induction of transcription of these genes. Furthermore, β-galactosidase staining, which is a marker of the SASP, was also performed on HDM-challenged nasal epithelial cells with or without ELVN-34 [60]. Treatment with ELVN-34 showed a marked decrease in β-galactosidase compared to cells treated with HDM alone.

In summary, our results show that ELVN-34 promotes cell viability and counteracts cytotoxicity upon HDM sensitization of HNEpC. Moreover, the levels of IL-6, IL-1β, IL-8, VEGF, ICAM1, CXCL1, and CCL2 increased upon HDM sensitization, which was reduced significantly upon ELVN-34 treatment. IL-10 level was reduced upon HDM sensitization, which was restored by ELVN-34 treatment. Our data also show that HDM triggered the expression of several key genes in the nasal epithelium, including autophagy, unfolded protein response, matrix metalloproteinases, senescence, and inflammation [30]. HDM activates the autophagy genes: ATG3, ATG5, ATG7, BECLIN-1, and P62. These data support other previous studies in other respiratory diseases as well as in COPD [34,53]. HDM treatment also increases the expression of UPR, including Atf6, Chop, and Ire1 genes, which are critical in allergic reactions and asthmatic patients [35,55,61].

In addition, the inflammatory genes IL-1α, IL-1β, and IL-6 are activated as well, and finally, the HDM also triggered senescence genes, including Cdkn1a (p21), Cdkn2a (p16), and Cdkn1b (p27), but not p53. Interestingly, ELVN-34 treatment diminishes the induction of the transcription of these genes. Moreover, β-galactosidase staining, a marker of SASP, was also performed in the nasal epithelial cells challenged with HDM with or without ELVN-34 [60]. ELVN-34 treatment showed a marked decrease in β-galactosidase when compared with cells treated with HDM alone.

Although more studies are needed, our results show that ELVN-34 are capable of inhibiting cytokines derived from epithelial cells and reducing the activation signal of the innate immune system by HDM. These activities are necessary in order to maintain adaptive immunity, restore the epithelial mucosal barrier, and reduce the transcription of senescence and autophagy genes. Altogether, this protective mechanism of action described for the first time for ELV could constitute a novel therapeutic option in the prevention and treatment of HDM allergy.

Allergens are associated with the SASP program, the inflammaging phenomenon, and senescent cells as a consequence of immuno-inflammatory dysfunction that leads to increases in oxidative stress, mitochondrial dysfunction, and the secretion of cytokines that contribute to cellular damage and chronic inflammation.

The integrity of barrier function in the nasal epithelium is very important for defense against pathogens, and therefore, impairment of barrier function might initiate and promote allergen sensitization and inflammation. In future experiments we should address the role that ELV play in maintaining the integrity of the epithelial cell barrier by determining the expression of different components of the tight junctions (claudin-1, -4, -7, -8, -12, -13, and -14, tricellulin, occludin, ZO-1, ZO-2, and JAM) on HNEpC.

ELV represent a new potential therapeutic supported by the data provided on their activity counteracting allergen-induced SASP, inflammaging, matrix remodeling, autophagy, and accumulation of senescent cells, thus restoring immune-inflammatory dysfunction and regulating homeostasis in the prevention and progression of allergic pathologies.

## Figures and Tables

**Figure 1 pharmaceutics-14-00113-f001:**
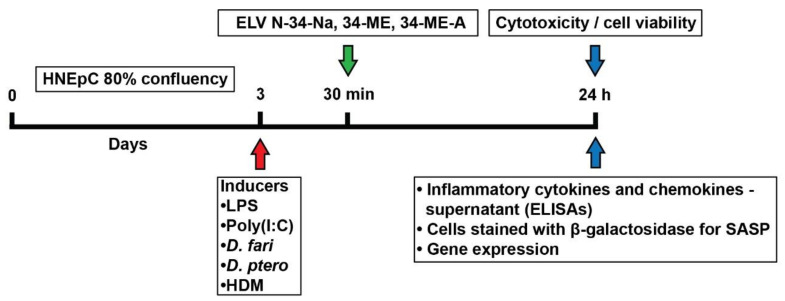
Experimental design. Human nasal epithelial cells (HNEpC) in primary culture challenged by allergy inducers—LPS, poly (I:C), D. fari, D. ptero, or a combination of house dust mite extracts (HDM) (*D. farinae* + *D. pteronyssinus*) (30 µg/mL) were analyzed to determine the protective effect of elovanoids (ELV)—ELVN-34-Na, 34-ME, or 34-ME-A (500 nM).

**Figure 2 pharmaceutics-14-00113-f002:**
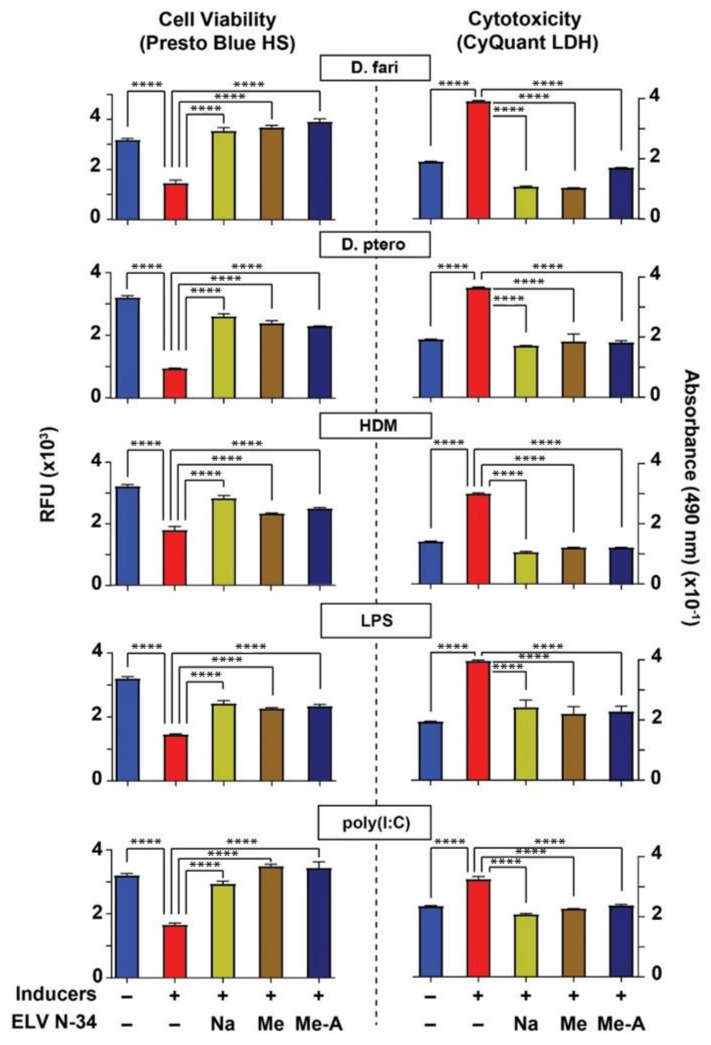
ELVN-34 protect HNEpC from cytotoxicity induced by HDM. Cell viability assay using Presto Blue HS kit (left panel) indicated that HNEpC treated with ELVN-34 (500 nM) were protected from cytotoxicity and cell death induced by *Dermatophagoides farinae*, *Dermatophagoides pteronyssinus*, and (*D. farinae* + *D. pteronyssinus*) (HDM), or LPS or poly(I:C) (30 μg/mL). Cytotoxicity assay using CyQuant LDH (right panel) shows that ELVN-34 (500 nM) elicit potent cytoprotection to HNEpC when challenged by the inducers. The results showed the averages of three independent experiments (**** *p* < 0.0001).

**Figure 3 pharmaceutics-14-00113-f003:**
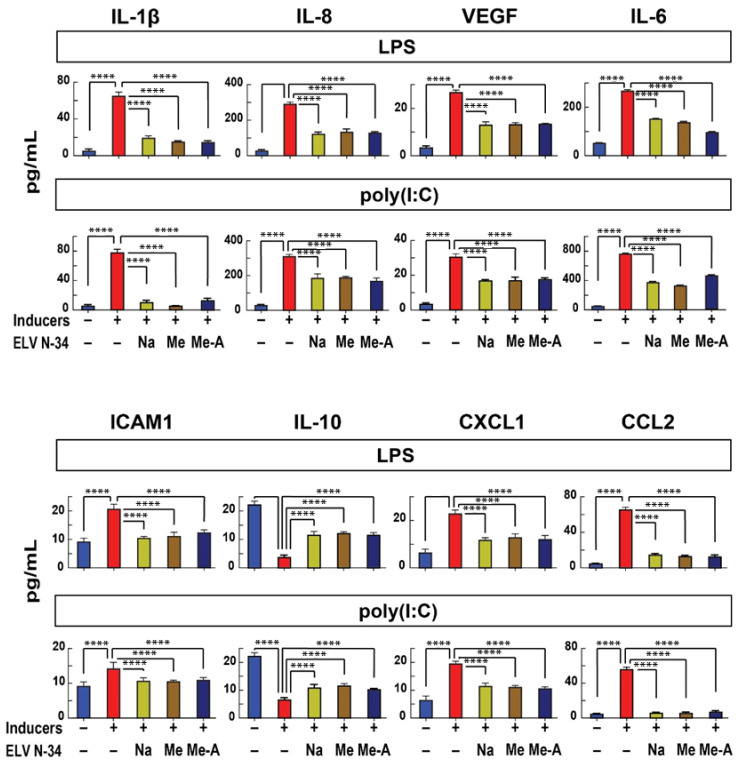
ELVN-34 reduce the expression of pro-inflammatory cytokines, chemokines and cell adhesion molecule in HNEpC challenged with LPS or poly(I:C). HNEpC challenged with LPS or poly(I:C) (30 μg/mL) display enhanced production of pro-inflammatory cytokines and chemokines IL-6, IL-1β, IL-8/CXCL8, CCL2/MCP-1, CXCL1/KC/GRO, VEGF, and cell adhesion molecule ICAM1(CD54) compared to non-treated cells. This increase in the production of pro-inflammatory molecules is abrogated by the addition of ELVN-34 (500 nM) 30 min post-challenge. LPS and poly(I:C) induce a reduction of anti-inflammatory IL-10 expression in HNEpC that is restored to normal levels after treatment with ELVN-34 (500 nM). The results showed the averages of three independent experiments. (**** *p* < 0.0001, NS—not significant).

**Figure 4 pharmaceutics-14-00113-f004:**
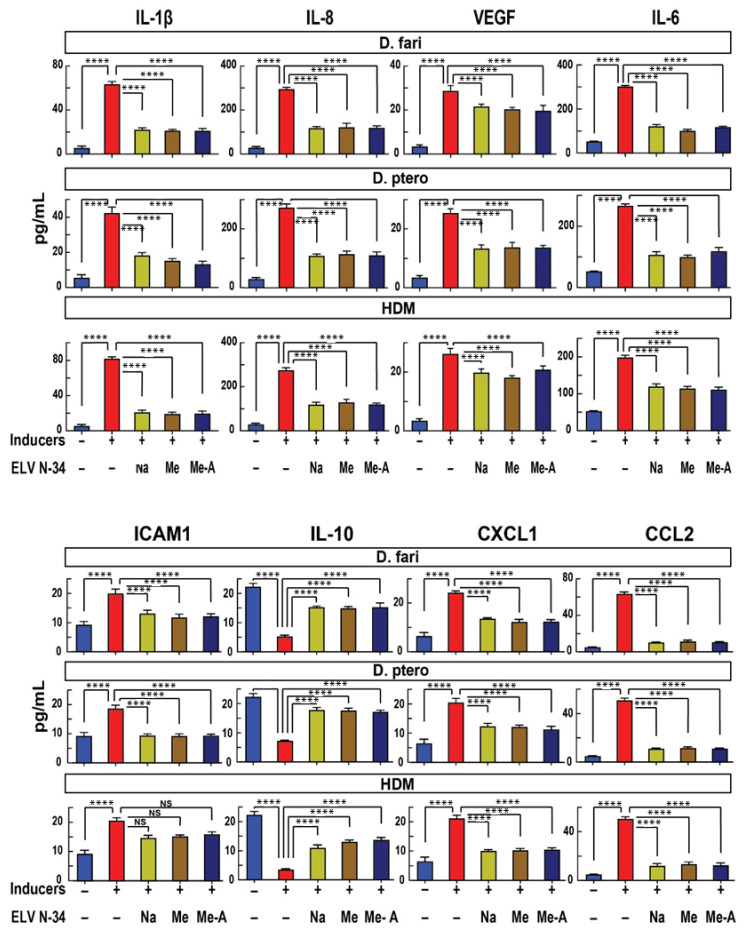
ELVN-34 reduce the expression of pro-inflammatory cytokines, chemokines, and cell adhesion molecule in HNEpC challenged with *D. farinae*, *D. pteronyssinus*, and HDM (*D. fari* + *D. ptero*) (30 μg/mL). ELVN-34 reduce the expression of pro-inflammatory cytokines, chemokines, and cell adhesion molecule ICAM1 in HNEpC exposed to HDM inducers. A remarkable decrease was observed in the release of anti-inflammatory cytokine IL-10 in HNEpC challenged with the HDM stressors (*D. farinae*, *D. pteronyssinus*, and HDM) (30 μg/mL). This decreased production of anti-inflammatory cytokine is reversed by the addition of ELVN-34 (500 nM), 30 min post-challenge with the stressor. The results showed the averages of three independent experiments. (**** *p* < 0.0001).

**Figure 5 pharmaceutics-14-00113-f005:**
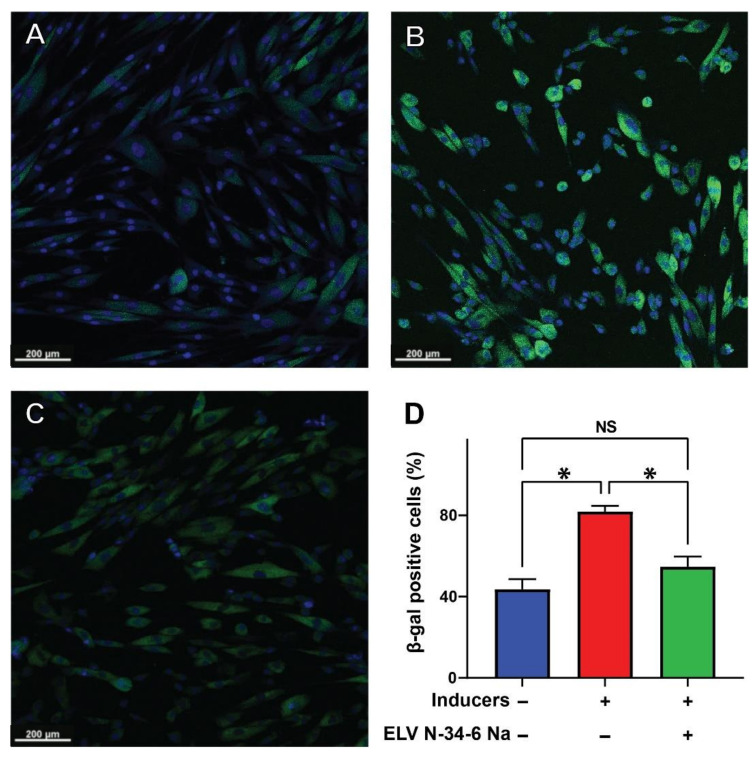
ELVN-34 protect HNEpC from the senescence-associated secretory phenotype (SASP response)**.** Representative images of HNEpC stained with Hoechst 33342 and spider β-galactosidase—(**A**) untreated (control), (**B**) challenged with HDM (30 μg/mL), and (**C**) HNEpC treated with HDM and ELVN-34:6 Na (500 nM). (**D**) Quantitative assessment of increased SASP response (measured by the spider β-galactosidase staining) upon HDM exposure (30 μg/mL). ELVN-34:6 Na (500 nM) treatment reduces the number of beta-gal-positive HNEpC induced by HDM. The results showed the averages of three independent experiments. (* *p* < 0.05, NS—not significant).

**Figure 6 pharmaceutics-14-00113-f006:**
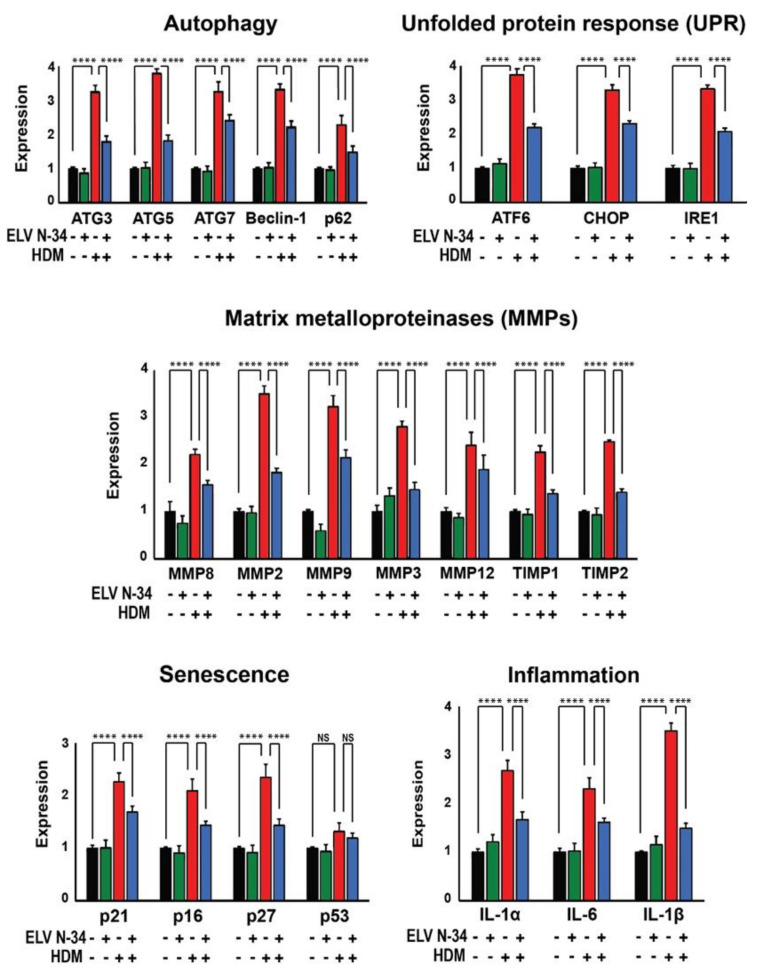
HDM triggered multiple signaling in HNEpC. HNEpC challenged with HDM (*D. farinae* + *D. pteronyssinus*) (30 μg/mL) induces the expression of several genes related with autophagy (ATG3, ATG5, ATG7, BECLIN-1, and P62), unfolded protein response (UPR) (ATF6, CHOP, and IRE1), and matrix metalloproteinases (MMPs) (MMP8, MMP2, MMP9, MMP3, MMP12, TIMP1, and TIMP2). HDM stressors also induce the expression of senescence (P21, P16, P27, and P53) and inflammation genes (IL-1α, IL-6, and IL-1β) on HNEpC. The treatment with ELVN-34 Na (500 nM) reduces the expression of autophagy, UPR, MMP, senescence (except P53), and inflammation genes induced by HDM extracts. The results showed the averages of three independent experiments. (**** *p* < 0.0001, NS—not significant).

## Data Availability

Not applicable.

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
