# Peer review of "Elovanoids Counteract Inflammatory Signaling, Autophagy, Endoplasmic Reticulum Stress, and Senescence Gene Programming in Human Nasal Epithelial Cells Exposed to Allergens"

_pharmaceutics, 2022, doi:10.3390/pharmaceutics14010113_

Round 1

Reviewer 1 Report

I have a question about the the experiment.

The concentration of LPS and polyIC seems to be too high. The authors should show the rationale for determining the concentration.

Do the cells used this time express TLR?

Is the production of IL-10 by epithelial cells? The cited literature is an abstract of an academic meeting and is not preferable as a citation of a paper. It is unlikely that the epithelium produces IL-10 without immune cells. This question is similar for other cytokines.

The measurement limits of ELISA should be indicated. Because of the low amount of cytokines shown, it is doubtful that the measurements are accurate.

Author Response

Journal:           Pharmaceutics (ISSN 1999-4923)

Title:                Elovanoids Counteract Inflammatory Signaling, Autophagy, Endoplasmic Reticulum Stress, and Senescence Gene Programming in Human Nasal Epithelial Cells Exposed to Allergens

Authors:          Alfredo Resano*, Surjyadipta Bhattacharjee, Miguel Barajas*, Khanh V. Do, Roberto Aguado-Jiménez, David Rodríguez, Ricardo Palacios, Nicolas G Bazan*

Manuscript ID:            pharmaceutics-1453084

We very much appreciate the prompt consideration of our manuscript and constructive criticism of the reviewer. Detailed below is a summary of the changes that have been made in the manuscript with appropriate rebuttal comments. The changes made on the manuscript addressing point-by-point according to the reviewer comments are itemized below. We hope that this improved manuscript satisfy the requests and is acceptable for publication in Pharmaceutics.

REVIEWER #1

I have a question about the experiment. The concentration of LPS and polyIC seems to be too high. The authors should show the rationale for determining the concentration.

We appreciate the comment regarding the concentration of LPS and poly(I:C) (30 μg/ml) used in our experimental procedure. The reason we use these concentrations is based on previous experiments that we had performed in the HNEC cell line in which we tested a dose-response curve using different concentrations of LPS and poly(I:C). According to our preliminary results, the doses showed in the manuscript are the ones that offered the best response in the HNEC cells regarding the main variables measured.

It is worth noting that we are not the only laboratory that has used this dose, since there are other colleagues who have published similar ranges of doses of LPS to that shown in the manuscript under review (https://doi.org/10.1016/j.waojou.2020.100109). Similarly, there are other colleagues who have used similar doses of Poly(I:C) (DOI: 10.1371/journal.pone.0098239) or a combination of both LPS and Poly(I:C) (DOI: 10.1186/s13601-015-0086-3).

In conclusion, we believe that the LPS and poly(I:C) (30 μg/ml) used in our experimental procedure are in line with those published in the scientific literature of the research area.

Do the cells used this time express TLR?

It is an interesting question, since the experiments carried out are meaningless without knowing the answer. In this sense, experiments carried out some time ago by our laboratory and by other publications confirm the expression of TLR (TLR-3, -7, -8 & -9) in HNEC cells (DOI: 10.1371/journal.pone.0098239). It has been also described the expression of TLR-1-6 and TLR-9 in HNEC cells by other authors (DOI: 10.1186/s13601-015-0086-3). Claeys et al. showed constant expression of TLR2 and TLR4 in tissue biopsies from patients with nasal polyposis or chronic rhinosinusitis and in healthy individuals (DOI: 10.1034/j.1398-9995.2003.00180.x).

Is the production of IL-10 by epithelial cells? The cited literature is an abstract of an academic meeting and is not preferable as a citation of a paper. It is unlikely that the epithelium produces IL-10 without immune cells. This question is similar for other cytokines.

Definitely yes. The expression of IL-10 by nasal epithelial cells has been already described (DOI: 10.1111/j.1398-9995.2007.01419.x), suggesting that nasal epithelial IL-10-secretion could regulate allergic symptoms. The secretion of other cytokines, such as IL-1b, CCL-5, IL-8, IL-18 and IL-33 was also described in HNECs of AR donors (DOI: 10.1016/j.waojou.2020.100109). Also, it has been described the expression of IL-6 and GM-CSF by human nasal epithelial cells (DOI: 10.1371/journal.pone.0098239).

Thanks for your appreciation regarding the references cited regarding cytokine secreted by human nasal epithelial cells. We have modified the cited references to include those mentioned above.

The measurement limits of ELISA should be indicated. Because of the low amount of cytokines shown, it is doubtful that the measurements are accurate.

Thanks for your appreciation, here is the range of values detected by each of the ELISA tests used in the manuscript:

IL-6 (Catalog# 6802, Chondrex) [9 - 600 pg/mL]

IL-1β (Catalog# 6805, Chondrex) [4 - 250 pg/mL]

IL-8/CXCL8 (Catalog# D8000C, R&D Systems) [31 - 2,000 pg/mL]

VEGF (Catalog# 6810, Chondrex) [31 - 2,000 pg/mL)

ICAM1(CD54) (Catalog# ab100640, Abcam) [150 pg/ml – 20,000 pg/mL]

CXCL1/KC/GRO (Catalog# 6825, Chondrex) [16 - 1,000 pg/mL]

CCL2/MCP-1 (Catalog# 6821, Chondrex) [16 - 1,000 pg/mL]

IL-10 (Catalog# 6806, Chondrex) [8 - 500 pg/mL]

We will include this information in the material and methods section.

The range of values detected by each of the ELISA tests used in the manuscript is in accordance with that published in the literature. The observed levels cannot be considered low since at the cellular level these levels are even higher than expected. It is assumed that the secretion of these cytokines will influence other cells, like immune cells (antigen-presenting cells, lymphocytes, monocytes,…), that are located in the near proximity of the nasal epithelium, thus acting in a paracrine manner. For this action, the levels necessary for these cytokines to be functional are usually in the “ng/mL” range.

General comments and conclusion:

As you may have notice, one of the corresponding authors is Nicholas G. Bazan, a world expert on neurological diseases (brain ischemia, parkinson and epilepsy) and the molecular mechanism involved in the brain protection thought the rapid release of unesterified essential fatty acids (docosahexaenoic and arachidonic acids) from membranes through phospholipase A2. Dr. Nicholas G. Bazan is the inaugural founder of The Ernest C. and Yvette C. Villere Chair for Research in Retinal Degeneration (1984-present) and he is also the founding Director of the Neuroscience Center of Excellence at the School of Medicine, Louisiana State University Health New Orleans (USA). He is a leading researcher on the role of Elovanoids protecting retinal pigment epithelial cells (Bhattacharjee et al. Science Advances, 2017; DOI: 10.1126/sciadv.1700735) and photoreceptors (Jun et al., Nature Scientific Reports, 2017: DOI: 10.1038/s41598-017-05433-7).

Dr. Nicholas G. Bazan has also recently published an article that highlights the role of Elovanoids in the infection of SARS-CoV-2 on human primary alveoli cells in culture (DOI: 10.1038/s41598-021-91794-z). The results showed a reduction on the infection through the blockage of the signal of spike (S) protein found in SARS-CoV-2 infected cells together with the downregulation of the angiotensin-converting enzyme 2 (ACE2) and enhancing the expression of a set of protective proteins hindering cell surface virus binding and upregulating defensive proteins against lung damage, suggesting that the Elovanoids are able to curb viral infection. These findings open avenues for potential preventive and disease-modifiable therapeutic approaches for COVID-19 using Elovanoids.

In this same direction, the present manuscript shows for the first time the role of the Elovanoids in the protection of Human Nasal Epithelial Cells exposed to allergy inductors (Dermatophagoides farinae and Dermatophagoides pteronyssinus) through the activation various signaling pathways such as autophagy, endoplasmic reticulum stress, and senescence, being one of the most promising future treatments. In fact, patients will be recruited soon in order to start a clinical trial in which the role of Elovanoids in allergy will be tested. The results showed in the paper submitted to Pharmaceuticals are supported by his experience in this field of research.

We believe it is a good opportunity for Pharmaceutics journal to publish these original results regarding the role of Elovanoids in the protection of human nasal epithelial cells exposed to allergens and we have no doubt that this article will receive multiple citations.

Reviewer 2 Report

Line 92: Materials and Methods move to the next lline.

Please modify all p value as *p < 0.05; **p < 0.01; ***p < 0.001; ****p < 0.0001 in this manuscript.

Like line 262: D. farinae, D. pteronyssinus.....should be italic in this manuscript.

Author Response

Journal:           Pharmaceutics (ISSN 1999-4923)

Title:                Elovanoids Counteract Inflammatory Signaling, Autophagy, Endoplasmic Reticulum Stress, and Senescence Gene Programming in Human Nasal Epithelial Cells Exposed to Allergens

Authors:          Alfredo Resano*, Surjyadipta Bhattacharjee, Miguel Barajas*, Khanh V. Do, Roberto Aguado-Jiménez , David Rodríguez , Ricardo Palacios , Nicolas G Bazan*

Manuscript ID:            pharmaceutics-1453084

REVIEWER #2

We appreciate the constructive comments of the reviewer. In this sense, we have made changes to the final manuscript that include all the reviewer's suggestions that are detailed below:

Line 92: Materials and Methods move to the next line.

Thanks, we have modified this point in the new version of the manuscript.

Please modify all p value as *p < 0.05; **p < 0.01; ***p < 0.001; ****p < 0.0001 in this manuscript.

Thanks for your appreciation, we should have done it as the reviewer indicates. This nomenclature has been included in the Figures, as you can notice in the new version of the manuscript.

Like line 262: D. farinae, D. pteronyssinus.....should be italic in this manuscript.

Sorry, it was an editing error in the final version of the manuscript. This aspect has been modified in the new version of the manuscript.

General comments and conclusion:

As you may have notice, one of the corresponding authors is Nicholas G. Bazan, a world expert on neurological diseases (brain ischemia, parkinson and epilepsy) and the molecular mechanism involved in the brain protection thought the rapid release of unesterified essential fatty acids (docosahexaenoic and arachidonic acids) from membranes through phospholipase A2. Dr. Nicholas G. Bazan is the inaugural founder of The Ernest C. and Yvette C. Villere Chair for Research in Retinal Degeneration (1984-present) and he is also the founding Director of the Neuroscience Center of Excellence at the School of Medicine, Louisiana State University Health New Orleans (USA). He is a leading researcher on the role of Elovanoids protecting retinal pigment epithelial cells (Bhattacharjee et al. Science Advances, 2017; DOI: 10.1126/sciadv.1700735) and photoreceptors (Jun et al., Nature Scientific Reports, 2017: DOI: 10.1038/s41598-017-05433-7).

Dr. Nicholas G. Bazan has also recently published an article that highlights the role of Elovanoids in the infection of SARS-CoV-2 on human primary alveoli cells in culture (DOI: 10.1038/s41598-021-91794-z). The results showed a reduction on the infection through the blockage of the signal of spike (S) protein found in SARS-CoV-2 infected cells together with the downregulation of the angiotensin-converting enzyme 2 (ACE2) and enhancing the expression of a set of protective proteins hindering cell surface virus binding and upregulating defensive proteins against lung damage, suggesting that the Elovanoids are able to curb viral infection. These findings open avenues for potential preventive and disease-modifiable therapeutic approaches for COVID-19 using Elovanoids.

In this same direction, the present manuscript shows for the first time the role of the Elovanoids in the protection of Human Nasal Epithelial Cells exposed to allergy inductors (Dermatophagoides farinae and Dermatophagoides pteronyssinus) through the activation various signaling pathways such as autophagy, endoplasmic reticulum stress, and senescence, being one of the most promising future treatments. In fact, patients will be recruited soon in order to start a clinical trial in which the role of Elovanoids in allergy will be tested. The results showed in the paper submitted to Pharmaceuticals are supported by his experience in this field of research.

We believe it is a good opportunity for Pharmaceutics journal to publish these original results regarding the role of Elovanoids in the protection of human nasal epithelial cells exposed to allergens and we have no doubt that this article will receive multiple citations.

Finally, we would like to thank you for the time you put in reviewing our paper and look forward to meeting your expectations. We look forward to hearing from you in due time regarding our submission and to respond to any further questions and comments you may have.

Reviewer 3 Report

Comments to the author

In the present manuscript, Resano et al. demonstrated the potential of Elovanoids as an anti-allergic agent in Human Nasal Epithelial Cells using various signaling pathways such as autophagy, endoplasmic reticulum stress, and senescence.

However, there are some points that need to be addressed by the authors.

Major concern:

  • The authors only evaluated the anti-allergic efficacy of Elovanoids (EVs) in this study. If you want to increase the potential of EV as an anti-allergic drug, it is necessary to compare the efficacy with other common anti-allergic drugs.
  • Tight junctions (TJs) comprise of cell-cell adhesion complexes between epithelial cells required for epithelial barrier function. TJs disruption plays causative roles in the development of allergy through increased exposure of nasal tissues to environmental allergens. TJs in the epithelial cells consist of three primary constituents of transmembrane proteins namely occludin, claudin, and junctional adhesion molecules. If EVs have the potential to be developed as a treatment for allergic rhinitis, please evaluate the effect of EVs in allergen-mediated dysregulation of tight junction-related proteins.

Author Response

Journal:           Pharmaceutics (ISSN 1999-4923)

Title:                Elovanoids Counteract Inflammatory Signaling, Autophagy, Endoplasmic Reticulum Stress, and Senescence Gene Programming in Human Nasal Epithelial Cells Exposed to Allergens

Authors:          Alfredo Resano*, Surjyadipta Bhattacharjee, Miguel Barajas*, Khanh V. Do, Roberto Aguado-Jiménez, David Rodríguez, Ricardo Palacios, Nicolas G Bazan*

Manuscript ID:            pharmaceutics-1453084

We very much appreciate the prompt consideration of our manuscript and constructive criticism of the reviewer. Detailed below is a summary of the rebuttal comments. We apologize in advance, since we have tried to be able to carry out some of the experiments proposed by the reviewer but, due to unforeseen events in the logistics of reagent shipments as a result of the merchandise transport problems, we have not been able to meet some of the reviewer's requirements. However, we do not consider that this experiment diminishes the importance of the role of Elovanoids that is described for the first time in the manuscript regarding their potential as a protective agent in human nasal epithelial cells through different signaling pathways such as autophagy, endoplasmic reticulum stress, and senescence.

REVIEWER #3

In the present manuscript, Resano et al. demonstrated the potential of Elovanoids as an anti-allergic agent in Human Nasal Epithelial Cells using various signaling pathways such as autophagy, endoplasmic reticulum stress, and senescence.

However, there are some points that need to be addressed by the authors.

Major concern:

The authors only evaluated the anti-allergic efficacy of Elovanoids (EVs) in this study. If you want to increase the potential of EV as an anti-allergic drug, it is necessary to compare the efficacy with other common anti-allergic drugs.

We appreciate the comment regarding the comparison of the anti-allergic efficacy of Elovanoids vs. other common drugs used as anti-allergic. We agree that this is an extremely interesting research question, however it was not the focus of our work since the present manuscript aims to demonstrate for the first time that Elovanoids can play an important role in the protection of nasal epithelial cells. Once the knowledge about the action of Elovanoids in human nasal epithelium cells has been established, the next step will be to compare this effect with other drugs already used as anti-allergic drugs. Even so, we can argue that the myriad of activities triggered by Elovanoids aimed to protect the nasal epithelial cells (showed for first time in our manuscript) makes us to hypothesize that they could have an anti-allergic effect even more complete than the drugs used today. The reason to argue this is based on the fact that any anti-allergic drug has been shown to be capable of counteract inflammatory signaling pathways, autophagy, improving endoplasmic reticulum stress as well as reducing the expression of senescence genes in human nasal epithelial cells just as Elovanoids have shown.

Tight junctions (TJs) comprise of cell-cell adhesion complexes between epithelial cells required for epithelial barrier function. TJs disruption plays causative roles in the development of allergy through increased exposure of nasal tissues to environmental allergens. TJs in the epithelial cells consist of three primary constituents of transmembrane proteins namely occludin, claudin, and junctional adhesion molecules. If EVs have the potential to be developed as a treatment for allergic rhinitis, please evaluate the effect of EVs in allergen-mediated dysregulation of tight junction-related proteins.

We agree that the integrity of barrier function in the nasal epithelium is very important for defense against pathogens, and therefore, impairment of barrier function might initiate and promote allergen sensitization and inflammation. In this sense, it has been described the role of IL-10 regarding the endothelial junction integrity and barrier function (DOI: 10.1006/mvre.2000.2288). We have described the expression of IL-10 by nasal epithelial cells, suggesting that nasal epithelial IL-10-secretion could regulate allergic symptoms may be though the upregulation of tight junction components, as the reviewer suggests. Unfortunately, due to problems with international shipments we have not been able to receive the antibodies on time in order to carry out the planned immunohistochemistry. We apologize to the reviewer for this setback, however, we do not consider that this experiment diminishes the importance of the role of Elovanoids that is described for the first time in the manuscript regarding their potential protecting human nasal epithelial cells through various signaling pathways such as autophagy, endoplasmic reticulum stress, and senescence.

However, we consider of great importance the contribution made by the reviewer regarding the role of tight junctions in the development of asthma, we have included the following paragraph in the conclusions of the manuscript:

“The integrity of barrier function in the nasal epithelium is very important for defense against pathogens, and therefore, impairment of barrier function might initiate and promote allergen sensitization and inflammation. In future experiments we should address the role that ELV play in maintaining the integrity of the epithelial cell barrier by determining the expression of different components of the tight junctions (claudin-1, -4, -7, -8, -12, -13, and -14, tricellulin, occludin, ZO-1, ZO-2, and JAM) on HNEpC.”

We hope that this improved manuscript satisfy the requests and may be acceptable for publication in Pharmaceutics journal.

General comments and conclusion:

As you may have notice, one of the corresponding authors is Nicholas G. Bazan, a world expert on neurological diseases (brain ischemia, parkinson and epilepsy) and the molecular mechanism involved in the brain protection thought the rapid release of unesterified essential fatty acids (docosahexaenoic and arachidonic acids) from membranes through phospholipase A2. Dr. Nicholas G. Bazan is the inaugural founder of The Ernest C. and Yvette C. Villere Chair for Research in Retinal Degeneration (1984-present) and he is also the founding Director of the Neuroscience Center of Excellence at the School of Medicine, Louisiana State University Health New Orleans (USA). He is a leading researcher on the role of Elovanoids protecting retinal pigment epithelial cells (Bhattacharjee et al. Science Advances, 2017; DOI: 10.1126/sciadv.1700735) and photoreceptors (Jun et al., Nature Scientific Reports, 2017: DOI: 10.1038/s41598-017-05433-7).

Dr. Nicholas G. Bazan has also recently published an article that highlights the role of Elovanoids in the infection of SARS-CoV-2 on human primary alveoli cells in culture (DOI: 10.1038/s41598-021-91794-z). The results showed a reduction on the infection through the blockage of the signal of spike (S) protein found in SARS-CoV-2 infected cells together with the downregulation of the angiotensin-converting enzyme 2 (ACE2) and enhancing the expression of a set of protective proteins hindering cell surface virus binding and upregulating defensive proteins against lung damage, suggesting that the Elovanoids are able to curb viral infection. These findings open avenues for potential preventive and disease-modifiable therapeutic approaches for COVID-19 using Elovanoids.

In this same direction, the present manuscript shows for the first time the role of the Elovanoids in the protection of Human Nasal Epithelial Cells exposed to allergy inductors (Dermatophagoides farinae and Dermatophagoides pteronyssinus) through the activation various signaling pathways such as autophagy, endoplasmic reticulum stress, and senescence, being one of the most promising future treatments. In fact, patients will be recruited soon in order to start a clinical trial in which the role of Elovanoids in allergy will be tested. The results showed in the paper submitted to Pharmaceuticals are supported by his experience in this field of research.

We believe it is a good opportunity for Pharmaceutics journal to publish these original results regarding the role of Elovanoids in the protection of human nasal epithelial cells exposed to allergens and we have no doubt that this article will receive multiple citations.

Finally, we would like to thank you for the time you put in reviewing our paper and look forward to meeting your expectations. We look forward to hearing from you in due time regarding our submission and to respond to any further questions and comments you may have.

Round 2

Reviewer 1 Report

The paper has been improved and I think it can be accepted.